# Voltammetric Detection of Glucose—The Electrochemical Behavior of the Copper Oxide Materials with Well-Defined Facets

**DOI:** 10.3390/s22134783

**Published:** 2022-06-24

**Authors:** Anna Kusior

**Affiliations:** Faculty of Materials Science and Ceramics, AGH University of Science and Technology, al. A. Mickiewicza 30, 30-059 Kraków, Poland; akusior@agh.edu.pl

**Keywords:** Cu_2_O, crystal-like, electrochemical performance, glucose detection, facet-dependent

## Abstract

Cu_2_O nanomaterials with well-defined facets and uniform size were synthesized by a wet-chemical method. Regardless of the additive composition, powders crystallize mostly in cuprite form. To compare their electrochemical behavior, the obtained materials were deposited on carbon glassy electrodes. The response to glucose from the materials with different exposed facets was recorded with a delay at the anodic curve. The chronoamperometric analyses (AMP) exhibited a lower signal in contrast to the cyclic voltammetry data (CV), indicating that the number of active sites involved in glucose oxidation processes resulting from the structure of the material is insufficient. For samples with dominant (100) or (111) planes, a typical characteristic was observed, however, with an additional peak at the anodic curve. The location of the peaks is approximately the same and no significant differences from the AMP and CV analysis were observed. The sample enclosed by the (111) facets exhibited higher activity; however, as a result of the redox reaction with glucose molecules, the surface state is changing. Cu_2_O materials enclosed by (100) planes exhibited optimal sensitivity as well as a large detective range. Samples with differential facet exposition present various current–potential profiles, as the effect of binder–particle interaction with Nafion.

## 1. Introduction

Electrochemical sensors play a crucial role in the detection of various compounds due to their high sensitivity, lower detection limit, high selectivity, and easy operation. Their specificity is based on the biological receptor layer coupled with a transducer for signal evaluation. The most common biosensors involve biocatalytic elements such as an enzyme, which can recognize its specific analyte [1]. By the detection mechanism, sensors were summarized into three generations: first sensors that exploit oxygen as a mediator, second employ artificial mediators, and third involve no mediator and direct electron transfer [2]. However, a problematic issue is a fact that enzymes are very delicate and prone to degradation when the temperature or pH values are not at optimum. Despite the new solutions used in the third generation of electrodes, it was found that the direct molecule oxidation at the electrode significantly slows reaction kinetics and increases the required potential. The solution was found to be in combination with basic nanomaterial properties: large specific surface area and tunable morphology. Therefore, interest has been focused on finding new materials [3], which were defined as fourth generation, based on nonenzymatic materials (inorganic). Due to intrinsic disadvantages, the last is considered the most promising.

The nonenzymatic receptor should fulfill the following requirements: high electrocatalytic activity, large surface area, effective electron transfer, good selectivity, high stability, and reproducibility [4]. Moreover, the direct oxidation and reduction processes on the nonenzymatic receptors turn out to be innovative and powerful. Various substances have been used to demonstrate their electroactivities, such as Pt, Au, paper, or carbon materials [5,6,7,8,9]. Despite a longer lifetime and inadequate sensitivity and selectivity, and lack of long-term stability, this makes efforts toward nanomaterials are based on transition compounds such as oxides or sulfides [10,11,12,13,14].

Among them, copper oxides (Cu_2_O, CuO) have attracted considerable attention. These p-type semiconductors possess unique optical and electrical properties. Moreover, due to their low cost, low toxicity, and handy synthesis method, they are extensively investigated in the field of catalysis, gas sensing, and photoelectrochemistry [15,16,17,18,19,20,21]. Due to the significant efficiency of the processes involved, more attention is being paid to copper oxide (I). One of the major properties of Cu_2_O materials is the possibility of showing different conductivity depending on the applied shape (existing grain boundaries) [15,22]. However, their electrochemical behavior is still not well understood [23]. 

The relation between microstructure and corresponding sensing properties may affect the development of the fourth generation of sensors due to the optimization of the mass and/or charge transfer [24]. Therefore, by enlarging the specific surface area, the number of active sites, as the main receptors, is expected to increase. The significant progress in materials science has led to a spectacular variety of building blocks of different sizes and shapes. Until now, structural defects, sometimes misunderstood as material flaws, have proven to be crucial in the field of photocatalytic processes. They not only act as an electron hole recombination place but also affect the electronic structure and take part in charge carriers traveling by acting as a scattering center [25]. Moreover, by implementing the defect engineering in the bulk, the light absorption range can be extended, while surface defects play a crucial role in the redox reaction at the surface (high energy and active sites) [26]. This raises the question of whether it is possible that defects previously regarded as ‘unwanted’ can improve the performance of electrochemical sensors. The importance of a better understanding of interface properties as a fundamental condition of further progress in materials science was highlighted. Particular attention should be paid to the fact that many functional characteristics of materials are determined by the interfacial layers (surfaces and grain boundaries) rather than by the bulk phases. 

One of the strategies involves the formation of materials with well-defined facets. The use of appropriately positioned atoms forming the whole network will bring many benefits. Mostly crystal-like structures are used in catalytic and photocatalytic processes. The use of naturally unsaturated bonds, which can be considered surface defects, increases the probability of enhancing the adsorption capacity of materials. Moreover, various crystallographic planes present different densities and atoms’ symmetry, which affect surface chemistry (energy, reactivity, and binding) as well as the possible changes in electronic structure [27,28,29]. The stability and properties of the materials are governed. The preferred growth direction is modified by the addition of the surfactant during the synthesis. Unfortunately, their use is associated with the possibility of leaving on their surface residues of used substances and thus blocking the active centers.

Electrochemical detection/biosensing of Cu2O showed that its variable performance depends on {111} or {100} facets. It is being claimed that Cu_2_O {100} possesses more neutral states than positively charged Cu_2_O {111} [30]. Xie et al. find out that lower activation energy is expressed by (100) planes. Moreover, due to the higher oxidation energy of the holes, low-work-function Cu_2_O (100) presented better sensing properties [31]. On the other hand, Wang et al. reported that {111} crystal planes display excellent sensing properties [32]. It can be concluded that the inferior electronic conductivity, which strongly affects the behavior of metal oxides, may be a restriction for their application. Therefore, the exploration of facet-dependent mechanisms is still investigated.

In this work, the Cu_2_O nanomaterials with well-defined facets and uniform size were synthesized by a wet-chemical method, however, with the addition of various surfactants. To compare their electrochemical behavior, obtained materials were deposited on carbon glassy electrodes. By analyzing cyclic voltammograms and amperometric signals, the effect of exposed planes was investigated. This work provides a valuable reference for the optimization of copper-oxide-based nonenzymatic sensors.

## 2. Materials and Methods

### 2.1. Synthesis of Cu_2_O Structures

Copper oxide materials were synthesized by adding to 200 mL distilled water 1.023 g of copper chloride dihydrate (CuCl_2_ × 2H_2_O, Avantor, Gliwice, Poland), 0.6 g of sodium citrate (C_6_H_7_NaO_7_, Sigma Aldrich, St. Louis, MI, USA), and 0.67 g of surfactant. To modify the shape of the received materials the following organic compounds were used: polyvinylpyrrolidone (PVP, M.W. 40000, Alfa Aesar, Haverhill, MA, USA), pluronic F (PF127, Sigma Aldrich, USA), polyvinyl alcohol (PVA, Sigma Aldrich, USA), sodium dodecylbenzene sulfonate (SDBS, Sigma Aldrich, USA), and ethylene glycol (EG, Avantor, Poland). The as-prepared solution was stirred for 10 min, followed by adding 20 mL of 6 M NaOH (Avantor, Poland). After 30 min, the 20 mL of 12 M l-ascorbic acid (AA, Avantor, Poland) was added dropwise to the solution and left for 3 h with continuous stirring. The whole process was carried out in a water bath at 60 °C. Details of the samples’ synthesis parameters are summarized in Table 1.

### 2.2. Surface Sensibilization by the Au Particles

To determine the plane with the highest surface energy, selective adsorption process was performed. Hierarchical heterostructures of Cu_2_O@Au were obtained by dissolution of the 0.3 g copper oxide powder in 30 mL of deionized water with (Au-PVP-CC1) addition of the 3 mL 1% PVP solution [33]. After 30 min of continuous stirring at room temperature, 1 mL 5 mM HAuCl_4_ aqueous solution was added (Sigma Aldrich, USA). The reaction time was 3 h. As-prepared samples were centrifugated at 6000 rpm for 5 min and washed 3 times with a mixture of deionized water and ethanol (50%/50%, *v*/*v*). 

### 2.3. Electrode Preparation

The carbon glassy electrode (GCE, 3 mm, Mineral Company, Warszawa, Poland) was polished with 0.1 and 0.3 μm Al_2_O_3_ powder (Avantor, Poland), respectively, rinsed with ethanol and distilled water. Electrochemical cleaning consisted of placing the electrode in 0.1 M H_2_SO_4_ (Avantor, Poland) solution in a three-electrode cell, with GCE, Pt wire, and Ag/AgCl electrodes (Mineral Company, Poland), and applying a constant value of 2000 mV and −1000 mV for 30 and 10 s, respectively. An amount of 5 mg of semiconducting powder was dissolved in an ultrasonic bath, in a suspension of isopropyl alcohol (Avantor, Poland), distilled water, and Nafion (Sigma Aldrich, USA) in a volume ratio of 2:7:1. Then, 10 μL of the mixture was coated on a cleaned GCE and dried in air at room temperature.

### 2.4. Characterization Techniques

The crystal structure and phase composition of obtained powders were analyzed using the X-ray diffraction technique (X’Pert MPD diffractometer, Phillips) and Raman spectroscopy in the range of 50–1000 cm^−1^ (Witek Alpha 300 M+ spectrometer) using 633 nm diode laser and 1800 diffraction grating. The morphology of copper oxides was carried out with the Nova NanoSEM 200 scanning electron microscope (FEI) equipped with a Helix detector. The electrochemical experiments (cyclic voltammetry, CV, and chronoamperometry, CA) were carried out with an M161 electrochemical analyzer (MTM-ANKO) with a conventional three-electrode cell, the unmodified and modified carbon glassy electrode (working electrode), Pt wire (counter electrode), and Ag/AgCl (3.0 M KCl, reference electrode). All of the measurements were performed in the 0.1 M NaOH under continuous stirring at room temperature. Voltammograms were recorded with a scan rate of 50 mV/s in the range of −200 to 1200 mV.

## 3. Results and Discussion

### 3.1. The Formation Mechanism of Variously Shaped Copper Oxides 

To investigate the formation mechanism of Cu_2_O particles with well-defined facets, various surfactants were applied. Figure 1 shows the SEM images of all the synthesized samples. 

The polyhedral architecture of CC1-CC4 samples contains high indexed planes (100), (111), (011), and (110), however, in a different ratio. The most balanced facet distribution exhibits CC1, while (100) is the dominant one for CC2-CC4. It is worth highlighting that CC2 has no presence of (111) planes, and CC5 polyhedrals are enclosed by (111) planes. Therefore, by using the ImageJ software, the area of the facet and area ratio of various facets were calculated. The obtained results are collected in Table 2.

Taking into consideration fact that the equilibrium form of the particle tends to possess minimal total surface energy, the surfaces with the lower energy should dominate the crystal. However, due to the preferential adsorption of additives on certain crystallographic planes, this order may be disturbed. The role of PVP in the shape evolution of cuprite oxide was investigated by Zhang et al. [34]. They proved that polarized functional group ‘-C=O’ easily interacts with the unsaturated Cu and thus stabilizes the crystal. The adsorption at the {111} is preferential and results in the inhibition of the growth of perpendicular planes. Herein, the applied amount of the PVP allowed for obtaining a system with dominant (100) and (011) planes (CC1, Table 2). 

While PVP contains proton-accepting carbonyl moiety, the PVA presents hydroxyl groups (CC5). Gu et al. showed that due to the presence of the saturated copper atoms, the {100} facets are more thermodynamically stable [35]. However, it was also found that the dandling bonds of unsaturated Cu atoms interact strongly with OH^-^ radicals. Due to the formation of hydrogen-bonding interaction, the growth mechanism of the particle occurs in one direction (111). On the other hand, according to the Sharma et al., the surface energy of the (111) plane is lower than that for (100) [36]. The transformation from {100} to {111} stable facet may be stopped by incorporation in the system growth-determining factors. It may be assumed that application of the same amount of the PF127 (CC2), SDBS (CC3), and EG (CC4) affects their adsorption at the ‘O’-terminated planes, thereby reduction in the (111) plane.

SEM images show that despite using various surfactant grain sizes for all samples, they are approximately the same. Therefore, to determine the adsorption capacity and define the plane with the highest surface energy of the materials, a simple experiment was performed. A sample with an almost uniform proportion of the planes (CC1) was firstly immersed in the PVP solution. Due to the ability of the adsorption at the surfaces with the highest surface energy, the polymer will block these active sites. Therefore, gold particles will adsorb on the less active sites, planes with a high level of free binding bonds, and would be distinct evidence, whereas (111) or (100) will be more responsible for the sensing properties of the materials. 

The SEM image of the obtained sensitized material is presented in Figure 2. As expected, the polymers’ presence on the surfaces with the highest surface energy affects the adsorption of the gold particle at the (100) planes. Results are in good accordance with the reported data [37]. Under the provided results, particles with the highest proportion (111) can be expected to be the most active concerning the adsorption of biomolecules.

Figure 3 represents the structural phase identification via XRD and Raman spectroscopy of the as-synthesized samples. The XRD patterns of the analyzed samples can be well indexed to the Cu_2_O (cuprite, JCPDS 01-078-2076). Only in the case of CC1 and CC2, a small amount of the CuO (tenorite, JCPDS 00-045-0937) is observed, with an amount of approx. 5 and 4%, respectively. The result shows that powders are highly uniform. No additional compounds were observed. The intensity ratio (γ) of Cu_2_O (111) and Cu_2_O (200) is the same for all analyzed powders, except for CC2 (Table 2). Such a notable distinction should be taken into account for further analyses of sensing properties.

Surprisingly, despite the similar grain size and phase composition, the development of the specific surface area and the shape adopted affect the observed color of the materials. Microphotographs of the powders taken under an optical microscope can be found in Figure 3a. It is well known that cuprite-based minerals show a reddish-orange color, while copper (II) oxide shows a brown/black color. Although data analysis suggests the presence of only 4% of tenorite and the same grain size comparable to the CC1 sample, implementation of the Pluronic F 127 results in obtaining dark powder.

Raman spectroscopy measurements (Figure 3b) were performed to further analyze the phase composition of the obtained materials. The vibrating bands reported for the crystalline structure of copper oxide (I) are observed at 106, 147, 217, 408, and 628 cm^−1^. The strongest mode at 217 cm^−1^ may be attributed to the 2Γ_12_ second overtones, while the peak at 411 cm^−1^ corresponds to Γ_15_ oxygen vacancies. The band at 147 cm^−1^ may be related to the not-active Raman mode generated only by local defects (Γ_15_ LO). The peaks at 106 and 628 cm^−1^ are assigned to the surface oxidation process of copper particles and 3Γ_12_ + Γ_25_ four-phonon mode, respectively. In contrast to the XRD data, small traces of CuO are observed in all obtained samples. The characteristic bands are observed at 294 (A_g_) and 611 cm^−1^ (B_g_) [38]. The performed analyses confirm that the applied capping agent makes it possible to obtain systems with dominant copper ions at a +2 oxidation state. 

### 3.2. Electroanalytical Characterization of the Copper Oxide Materials

The obtained copper oxide materials were tested as the receptors for glucose detection. Powders were deposited at the carbon glassy electrode using the drop-casting technique. Figure 4 shows images of prepared paste drops displaying powder dispersion in the detection layer. The most homogenous particle alignment is presented for CC5. In other cases, copper oxide accumulates at the edge of the droplet. It is worth highlighting that despite the powder aggregation, the single particles do not form large agglomerates. The higher-resolution microphotographs prove the formation of the interface between single copper oxide grains/small aggregates and GCE surfaces. It can significantly affect the quality of the received signal and the sensitivity of the system. However, the reproducibility test shows the recorded signal in the presence of glucose molecules, despite displaying the paste by drop; the recorded data are repeatable (Appendix A).

#### 3.2.1. Evaluating Chemically Reversible Electron Transfer

The modified GCE electrodes with copper oxide nanoparticles were firstly tested in a chemically reversible ferrocyanide/ferricyanide solution to investigate the electrochemically active surface of the working electrode. Current–voltage characteristics were recorded for various scan rates in the range from 12.5 to 1000 mV/s. Results were collected in Figure 5, Figure 6 and Figure 7 and Table 3.

From the observation of the shape of the voltammograms, the analyzed materials can be divided into two groups. In the first one, copper oxide particles are characterized by the presence of variously oriented planes (CC1–CC3), while in the second one, the materials have one dominant plane (CC4, CC5). 

In Figure 5a–c, two redox pairs/reactions can be observed. When the electrode is scanned in a positive direction, due to the oxidation process an anodic current appears (i_pa_). Depending on the proportion of individual planes (Table 2), first, oxidation of the redox-active species occurs at 320, 380, and 200 mV vs. Ag/AgCl for the CC1, CC2, and CC3, respectively. This peak is hardly visible and its shape is characteristic of microelectrodes. The second reaction seems to be more intensive and faster. Oxidation begins depending on a sample at 755, 590, and 736 mV. It is worth highlighting that the electrode voltammetric response is typical for microelectrodes.

After the sweep of the potential and scanning in the opposite direction, due to reduction processes cathodic current is generated (i_pc_). At the voltammograms, two processes were recorded, which can be easily assigned to copper oxide reduction (720 and 710 mV) and [Fe(CN)_6_]^3−^ ion formation (230, 250 mV). Both the height of the peaks and their proportionality (i_pa_/i_pc_ ≈ 1) demonstrate that the observed processes are chemically reversible and based on electron transfer. The characteristic trumped plot of the peak current as a function of the square root of the scan rate ν^0.5^ is shown in Figure 5d–f. In both cases with the increasing scan rate, the shift towards higher potential values is visible. It is worth noting that the signal coming from the redox reaction of Cu^1+/2+^ (red squares) is much larger than for [Fe(CN)_6_]^3−^ (black circles). By analyzing the slope of the linear fit of the cathodic peak for both groups, where copper ions yield values of 4.27 × 10^−4^ (R^2^ = 0.9983), 3.27 × 10^−4^ (R^2^ = 0.9870), 4.61 × 10^−4^ (R^2^ = 0.9961), and ferrocyanide 1.20 × 10^−4^ (R^2^ = 0.9225), 3.04 × 10^−5^ (R^2^ = 0.9823), 6.01 × 10^−5^ (R^2^ = 0.680) [AV^−0.5^] for CC1, CC2, CC3, respectively, it can be assumed that the diffusion of copper ions affects the surface state.

The response of the modified GCE electrode by CC4 and CC5 powders is different than for samples CC1–CC3 (Figure 6a,b). Firstly, CC4 easily supports the redox reaction of the ferrocyanide–ferricyanide coupled system. The signal is sharp, and waves appear in the symmetric form, with similar peak potential. Moreover, the oxidation of the Cu^1+^ also takes place. Analysis of the cathodic and anodic peak values (Figure 6c) suggests that reactions involving the transition of Cu^1+^ to Cu^2+^ are an irreversible process (i_pa_/i_pc_ ≠ 1). As a consequence, the condition of the active centers is being changed. A slightly different situation can be observed for sample CC5, where planes (111) dominate. The irreversibility of the redox reaction of the copper ions (Figure 6d) is also visible. However, the apparent lack of symmetry of the currents for the ferrocyanide/ferricyanide system is surprising. The observed changes may indicate that the surface of the material (distribution and type of active centers) favors their reduction processes.

To exclude the influence of the host material (pure GCE) and the polymer solution used as an adhesive, tests were performed in 0.1 M KCl solution, and with the addition of ferrocyanide–ferricyanide. The results are summarized in Figure 7a,b. In both cases, a signal for glassy carbon was recorded, but the magnitude of the signal is significantly lower in comparison to the response of oxide materials. Moreover, the application of polymer solution inhibits the effect; thus, it can be assumed that the observed reactions in Figure 5 and Figure 6 are the result of Cu_2_O and electrolyte interactions. To further analyze the occurring phenomenon, measurements were performed in a potassium chloride electrolyte and the results are summarized in Figure 7c. Each of the samples tested responded to the presence of KCl in the solution. Despite the low signal for CC1-CC3 materials, materials with a dominant single plane type are affected by the presence of the potassium and chloride ions. However, both the potential range and current values do not correspond to those obtained in the presence of [Fe(CN)_6_]^4−/3−^. Therefore, the observed peaks in Figure 5 and Figure 6 are not due to the KCl reaction. 

Therefore, it can be assumed that the [Fe(CN)_6_]^4−/3−^ will approach the working electrode where the reduction takes place; then, ferrocyanide ions also will diffuse into the solution. The recorded current is an appropriate measure of the overall reaction rate. The standard oxidation potential of the ferrocyanide–ferricyanide was determined at approx. 0.37 V vs. Ag/AgCl [39]. Therefore, it is assumed that peaks at the lower potential values can be assigned to the reversible ferrocyanide/ferricyanide coupled system. The second reaction which can take place is related to the copper oxide receptor. As a result of the adsorption of the [Fe(CN)_6_]^4−^ ions at the surface of copper oxide and applied potential, the oxidation of copper may occur. The reduction potential of copper oxides depends on the layer thickness and the presence of hydroxyl radicals. It was reported that copper (I) oxide undergoes reduction at a range from −0.5 to −0.8 V (vs. Ag/AgCl), while tenorite may be upgraded at +3 when 0.65 V (vs. Ag/AgCl) is reached [14,40,41]. The obtained samples consist of min. 94% of cuprite; therefore, the reaction Cu^1+^ → Cu^2+^ is the most probable and responsible for the formation signal at the voltammograms. The changes in the current as a square root of the scan rate for this reaction are shown in Figure 5d–f (red square points). 

On the other hand, the affinity of Cu^1+^ → Cu^2+^ oxidation for sample CC5 is slightly higher; however, the signal of i_cat_ suggests reduction processes are determining the electrode response. Not only the copper (I) oxide reaction is irreversible, but also, due to changes in the powders’ surfaces, the [Fe(CN)_6_]^4−/3−^ will occur only in one direction. While the (100) planes are reported to be less stable than (111) or (110) [37,42], the electrochemical behavior of CC5 is unusual.

Based on the obtained results denoted to the ferrocyanide–ferricyanide coupled system, and assuming that the bulk concentration is constant and these reactions are diffusion-controlled, the Randles–Sevcik equation was used to determine the electrochemically active surface area, EAS [43]. The results are collected in Table 3. CC4 is characterized by the highest EAS, while CC3 is the smallest. For sample CC5, due to the irreversibility of the process, EAS could not be determined.

#### 3.2.2. Catalytic Currents

The obtained materials with defined planes were subjected to electrochemical tests against the presence of glucose. The results are summarized in Figure 8 and Figure 9. 

The modified electrodes were analyzed from −50 to 1200 mV, with a scan rate of 50 mV/s. The measurement was performed in a 0.1 M NaOH medium. Due to the response of the modified electrodes, they were divided into two groups, similar to the ferrocyanide–ferricyanide assays.

First of all, for samples CC1 to CC3, a signal was obtained whose value increased with an increasing amount of glucose in the systems. Surprisingly, it appeared on the anodic curve (marked as 1). Analogous tests in a PBS environment and for smaller potential ranges did not reveal (up to 800 mV) an indication that the material was sensitive to the analyte added to the system. Moreover, analysis by implementation of the SVC technique does not outline the input of Faradaic current. Considering the previous behavior of the material in the presence of an additional redox couple, it can be assumed that only after a certain potential is exceeded that the copper ions excited, which allows the oxidation of glucose. Due to its high value, the response of the electrode to the presence of sugar is recorded with a delay, as an anode current. To take a closer look at this phenomenon, chronoamperometric measurements (at a fixed potential indicated by the anode peak, Appendix A) were performed. This measurement allowed finding out if indeed a higher potential difference is required to activate the copper ions than the value of the resulting signal on the CV curve would indicate. The summarized results are shown in the inset graphs in Figure 8a–c. For samples CC1 and CC2, the values of the generated currents at the CV curves were low (green triangles) and the range of response linearity was quite narrow. In comparison, a nearly 10-fold decrease in signal is noticeable in amperometric signal (black circles). This may indicate that the number of active sites involved in glucose oxidation processes resulting from the structure of the material itself is insufficient. The cyclic voltammetry allows for activation of the more active sites due to the stable increase in the potential. This type of measurement allows overcoming the barrier that allows the oxidation state of the compound to change, and the biomolecule detection process to occur. The different nature of the changes is shown in sample CC3. The generated current is significantly higher and increases linearly over a wide range of glucose concentrations. In addition, for this type of measurement (AMP), the shape of the resulting curve has a characteristic parabolic form. For cyclic voltammetry measurements, the shape of the curve obtained allows the two curves to match, which suggests that the reactions taking place on the surface of the material occur at different rates. A comparison of the slope for the CV (first stage) and AMP measurements are given in Table 3. For samples CC1–CC3, the slope values obtained from the two different types of measurements are markedly different with those from CV being significantly larger and closer together. The nature of these changes confirms the ‘activation’ of some of the active centers involved in the glucose detection in the process of linear increase in the applied potential, which in turn improves the sensor response. It is completely different for CC4 and CC5 powders, where these values oscillate at the same level. 

On the other hand, for materials obtained using PVA and EG, a typical characteristic curve can be observed (Figure 9a). The signal increases on the cathode curve with the amount of glucose added, and the value of the generated current is high (marked as 2 on the graph). At the voltammograms, also visible is the peak at the anodic curve. The location of the peaks is approximately the same. Moreover, the shift towards the higher values of potential is observed (Figure 9b) for both signals. 

Analysis of the i_pa_ and i_pc_ signal (CV measurements, Figure 9b) showed no significant differences from the amperometric measurements.

Based on the obtained data, the basic parameters characterizing the sensor systems were determined, including a limit of detection (LOD), the limit of quantification (LOQ), and sensitivity per unit (S) toward:LOD = 3.3 SDa^−1^(1)
LOQ = 10 SDa^−1^(2)
S = I(CA)^−1^(3)
where SD is a standard deviation, a—slope of linear regression, I—the peak current (µA), C—glucose concentration (mM), and A—active surface area; calculations were made towards EAS and carbon glassy surface area (Table 3). 

As can be concluded from Table 3, glucose can be detected from 0.07 mM and quantified from 0.2 mM when using CC4 (amperometric measurements) or CC3 (cyclic voltammetric data). However, due to the functionality and potential use of the materials as sensor receptors, better optimal properties are demonstrated by CC4. The sensitivity of the GCE-modified electrodes was evaluated by the electrochemically determined active surface area (EAS) and by the geometrical surface of the electrode. Higher values were obtained for calculations against the theoretical electrode area size. It is worth noting that they do not result from the actual available active sites available on the receptor; therefore, EAS measurement would have to be used for consideration and comparison. The smallest changes in the glucose-concentration-influenced signal were recorded for CC3, whereas the weakest one was for CC2.

## 4. Discussion

The development of fourth-generation sensor chips based on inorganic semiconductor materials is challenging due to the unknown matric effect of real samples. Therefore, the good sensing surfaces for analysis should fulfill the following requirements: inexpensive materials for surface modification, simplicity, and easiness of modification, chemical stability, reproducibility, capability to directly analyze the real samples, and simplicity in production. 

The sensing mechanism is based on conversion inversion processes at the inert surface with the presence of hydroxyl radicals. The basic electrode reaction involves mass and electron transfer. The first step in glucose detection of copper oxide materials is based on adsorption at the specific active centers. So far, the impact of surface engineering on the unique properties of the materials was mostly analyzed in the field of catalytic processes, especially photocatalysis and gas sensing. 

The increased efficiency of the processes, the wide range of sensitivity, and the longer lifetime are primarily influenced by surface development (smaller grain size) and the presence of planes with increased surface energy. Many papers are devoted to charge carrier separation and the role of (100) and (111) planes, which significantly may affect the copper oxide’s catalytic performance [19,44,45,46]. So far, Liu et al. proved that {111} planes are more active than other planes due to the presence of unsaturated copper atoms [47]. Moreover, the hexapod structure enclosed by these planes exhibits a wide detection range and the possibility of Cu^3+^ formation. However, this theory contradicts the results obtained in this paper. Comparing the samples CC4 and CC5, which are, as it were, enclosed by planes 100 and 111, respectively, a better glucose response was obtained for CC4. 

In contrast, according to Jiang et al., the {111} should firstly accelerate electron transfer [48]. However, the higher catalytic activity should be assigned to the {100} facets. They highlighted that the intensity ratios γ of copper oxide (200) and (111) planes may be evidence of the higher sensing/electrocatalytic activity due to the shifting toward lower potential values of initial glucose oxidation. In the present study, the γ is comparable with all samples despite the shape. However, the significant role of the (100) plane stayed in good correlation with Jiang et al.’s statement, as well as with DFT calculations [42]. A stronger interaction should be expected between glucose molecules and the {100} planes due to the significantly higher surface energy. Moreover, the presence of defects and relaxation would affect increasing these values. It is also well known that due to the dandling bonds, the surface energy activity follows (110) > (111) > (100) [49,50]. Taking it into consideration, it may be assumed that sample CC5 exhibits higher activity; however, as a result of the redox reaction with glucose molecules, the surface state is changing. Moreover, it is worth noting that for systems with different planes (CC1–CC3), i_pa_ analysis (CV measurements) showed the possibility of matching two curves, which may indicate the presence of different rates of glucose oxidation reactions (Figure 8). According to the calculations made from the SEM images (Table 2), the contribution of the (100) plane is dominant. Nevertheless, facets (110) and (011) have a significant percentage of a single particle, which suggests that they may be responsible for changing the kinetics of redox processes.

On the other hand, the interesting point is the electrochemical behavior of the CC1-CC3 copper oxide samples according to their chemical composition. The recorded signal responsible may be also the synergetic effect of the Cu_2_O-CuO (approximately 4% CuO), which positively affects the electron transport and surface redox reaction. One of the proposed mechanisms of glucose detection by Cu_2_O involves adsorbed hydroxyl molecules and electrochemically activated formation of Cu(OH)_2_ [51]. The electrocatalysis mechanism of the cuprite materials toward the glucose follows [48]:Cu_2_O + 2OH^-^ → 2CuO + H_2_O + 2e’(4)
CuO + OH^-^ → CuOOH_2_ + e’(5)
2CuOOH_2_ + glucose → 2CuO + gluconic acid + H_2_O(6)

It follows that for the process of glucose detection to take place, there must be a conversion of Cu^1+^ to Cu^2+^, then to Cu^3+^, and finally due to oxidation of glucose Cu^3+^→Cu^2+^. According to Jiang et al. [48], the analyte molecules should firstly be adsorbed at the {100} facets. Reactions 4 and 5 should take place easily taking into consideration the low energy activation and work function [31]. Moreover, the presence of more positively charged {111} facets should affect the amount of the adsorbed hydroxyl radical, which supports these reactions. In spite of this, the results showed that participation of the (100) plane is dominant for all samples (Table 2); the recorded signals occur at the anodic curve as delayed information from glucose oxidation. Gao et al. analyzed the impact of {100} and {111} and formed an interface on the electroreduction of ethylene [52]. They proved that reaction products can be hardly escaped from the Cu_2_O surface. In addition, the presence of the {111} should promote charge transfer between facets and, therefore, desorption of the analyte. Taking into consideration the first part of their studies, the electrocatalytic behavior of the copper oxides would be understandable, but contradictory to the results obtained from the sample CC4. 

The reason for this behavior is to be found in the stabilizer used, Nafion, rather than in the surface modifiers used, such as PVP or PVA [53,54]. Nafion is a charged polymer, which is expected to affect the double-layer structure, and therefore the charge of the particles. As a result, the effects caused by the (111) and (100) facets may not be observed. Assuming that the forced formation of specific planes and thus the arrangement of atoms on the surface is a deliberate introduction of defects into the system, it can be concluded that the polymer blocked these active centers thus preventing the adsorption of glucose molecules and hindering the Cu^1+^→Cu^2+^→Cu^3+^ processes. This effect can be observed by comparing measurements made using cyclic voltammetry and amperometry. If we scan the sample by gradually increasing the potential, the material can overcome the barrier that Nafion represents. If we use a fixed potential, the resulting signal is small. In contrast to the CC4 sample, these materials are characterized by a rather diverse array of planes with different energies. As a result of selective adsorption of Nafion on copper oxide grains, partial charge neutralization may occur.

For analyses of the mechanism by which surface engineering affects the catalytic ability of a material, the interface at the material/electrolyte interface and the properties of the material itself are considered. In the case of electrochemical sensors, it also appears that the influence of the linker used between the material and the conducting substrate is also crucial. This study aimed to determine whether it is possible to translate the rules of catalytic processes into mechanisms related to the electrochemical detection of biological substances. This work provides a valuable reference for the optimization of nonenzymatic sensors, not only highlighting the effect of (100) and (111) planes but also introducing the role of the binder. 

## 5. Conclusions

In summary, differently shaped copper oxides with well-defined facets were obtained. Depending on the applied surfactant, various facets may be presented. The applied amount of PVP allowed obtaining particles with balanced facet distribution, while PVA enclosed grains in a (111) polyhedral shape. Regardless of the additive composition, powders crystallize mostly in cuprite form. Cu_2_O materials enclosed by (100) planes exhibit optimal sensitivity as well as a large detective range. Samples with differential facet exposition present various current–potential profiles, due to the effect of binder–particle interaction with Nafion.

## Figures and Tables

**Figure 1 sensors-22-04783-f001:**
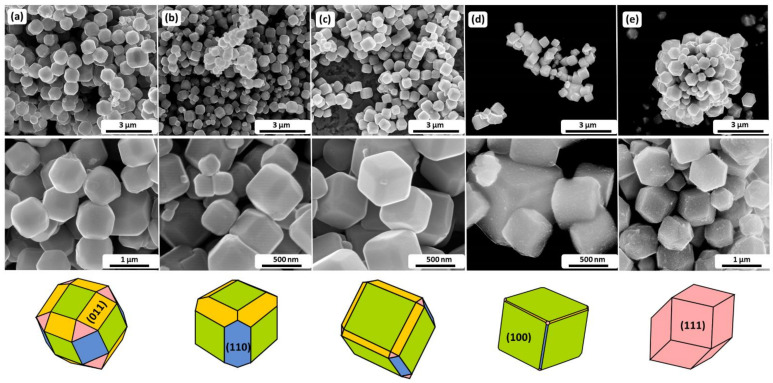
SEM images of the obtained polyhedral Cu_2_O particles with their schematic representation depending on the surfactant used: (**a**) PVP, (**b**) PF127, (**c**) SDBS, (**d**) EG and (**e**) PVA.

**Figure 2 sensors-22-04783-f002:**
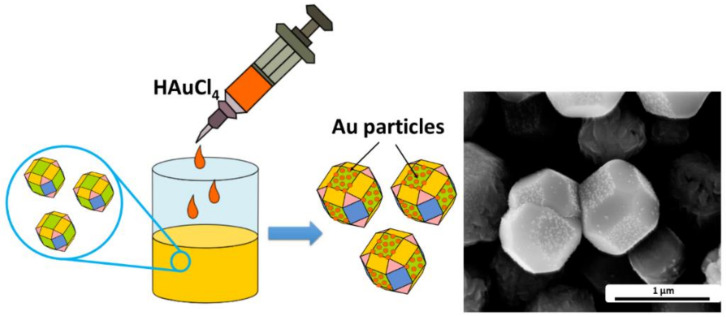
Schematic illustration of Au particle adsorption at the polyhedral surface with the SEM image of obtained modified copper oxide grains (CC1-Au).

**Figure 3 sensors-22-04783-f003:**
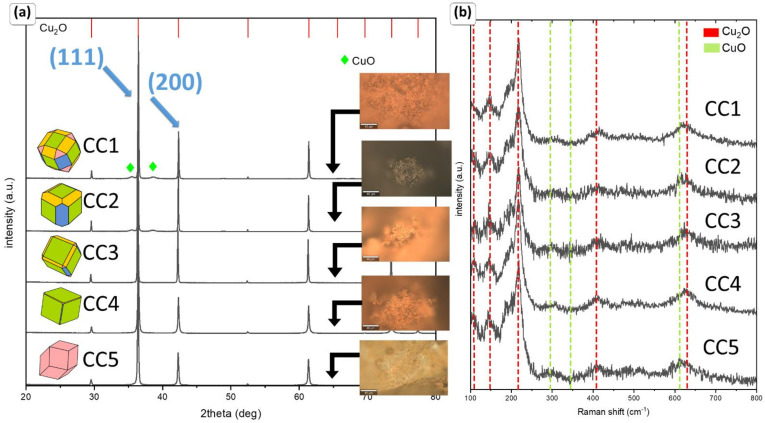
The XRD data analysis with microphotographs from optical microscopy (**a**) and Raman spectra of obtained polyhedral materials (**b**).

**Figure 4 sensors-22-04783-f004:**
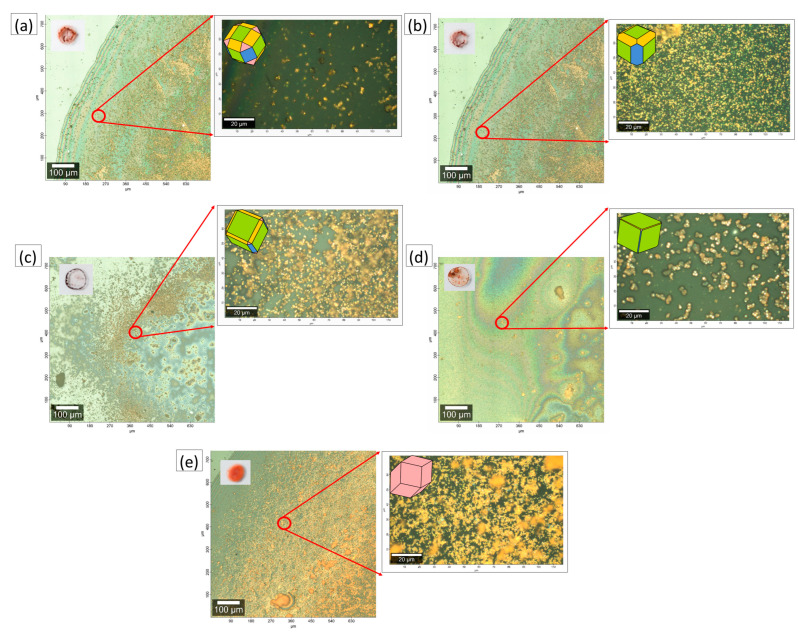
Microscope images of the dry deposits: (**a**) CC1, (**b**) CC2, (**c**) CC3, (**d**) CC4, and (**e**) CC5, with the photographs of prepared paste drops displaying powder dispersion in the detection layer (left upper corner).

**Figure 5 sensors-22-04783-f005:**
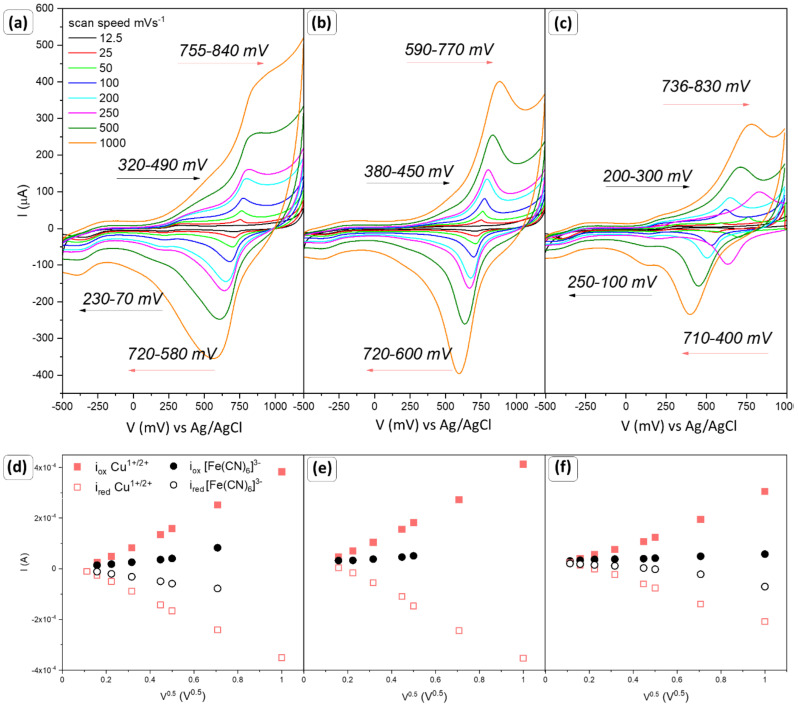
Cyclic voltammograms of a 1 mM [Fe(CN)_6_]^3-^/[Fe(CN)_6_]^4-^ in a 0.1 M KCl at different scan rates from 12.5 to 1000 mV/s for (**a**) CC1, (**b**) CC2, and (**c**) CC3, with a corresponding plot of the anodic and cathodic peak potentials (**d**–**f**) as a function of the square root of the scan rate ν^0.5^, respectively.

**Figure 6 sensors-22-04783-f006:**
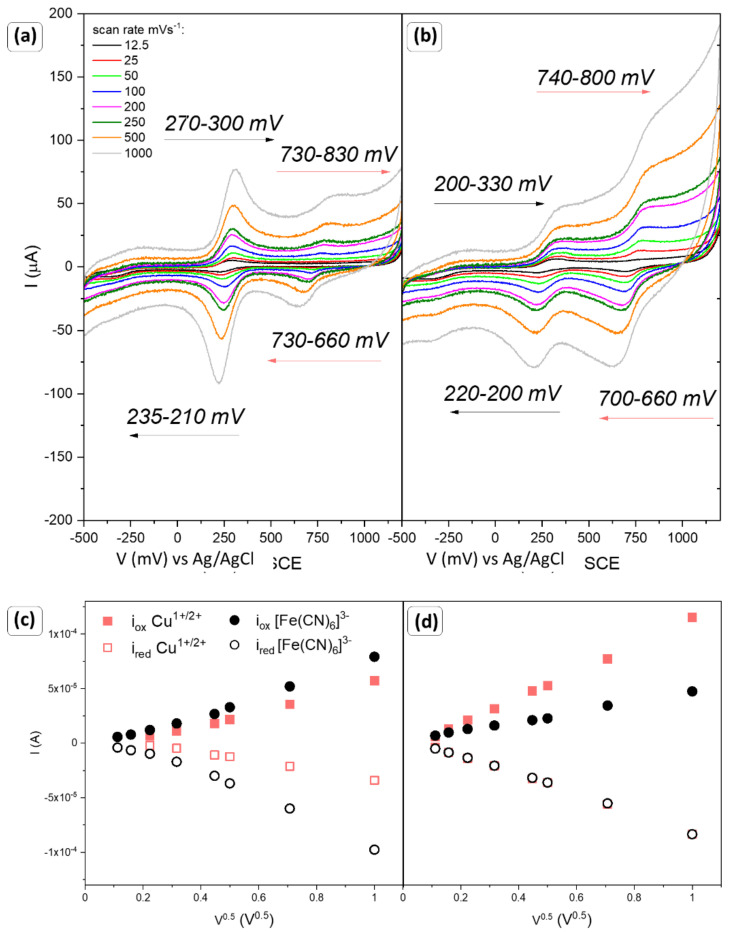
Cyclic voltammograms of a 1 mM [Fe(CN)_6_]^3−^/[Fe(CN)_6_]^4−^ in a 0.1 M KCl at different scan rates from 12.5 to 1000 mV/s for (**a**) CC4 and (**b**) CC5, with a corresponding plot of the anodic and cathodic peak potentials (**c**,**d**) as a function of the square root of the scan rate ν^0.5^, respectively.

**Figure 7 sensors-22-04783-f007:**
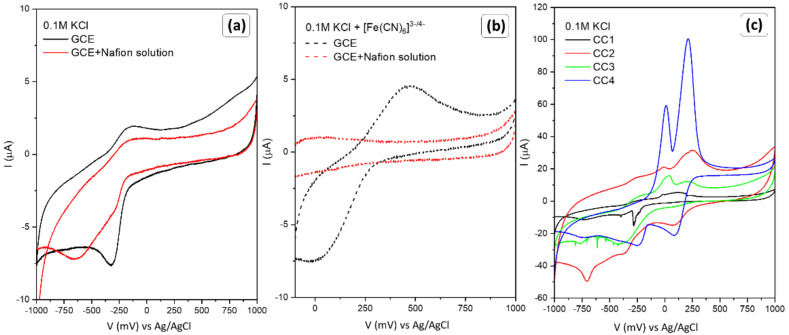
Cyclic voltammograms of GCE and GCE + Nafion solution electrodes in (**a**) 0.1 M KCl and (**b**) 0.1 M KCl with 1 mM [Fe(CN)_6_]^3−/4−^ at 50 mV/s. The response (**c**) of copper-oxide-modified electrodes towards the presence of the 0.1 M KCl.

**Figure 8 sensors-22-04783-f008:**
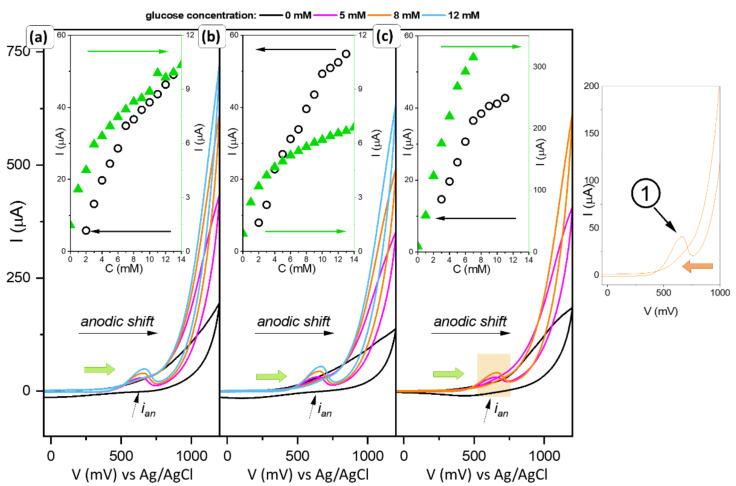
Electrochemical performance of the electrodes (**a**) CC1, (**b**) CC2, (**c**) CC3 powders, with the calibration curves (insets) of the glucose concentration vs. generated current, data obtained from the cyclic voltammetric (black circle point), and amperometric (green triangle points) measurements. On the right, the higher resolution of the formed peak is generated at the cathodic curve.

**Figure 9 sensors-22-04783-f009:**
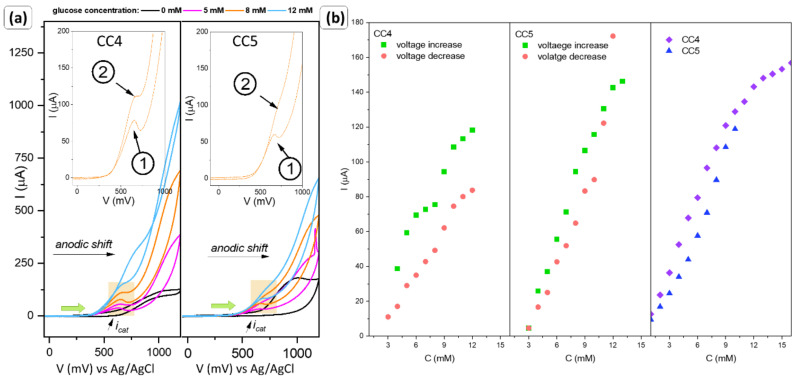
The electrochemical performance of the electrodes CC4 and CC3 powders, with the higher resolution of the formed peak generated at both the anodic and cathodic curves (**a**), and the calibration curves of the glucose concentration (**b**).

**Table 1 sensors-22-04783-t001:** Summary of materials’ synthesis parameters.

Sample	CC1	CC2	CC3	CC4	CC5
shape	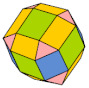	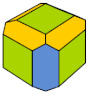	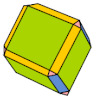	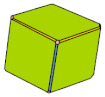	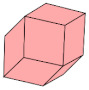
precursor	CuCl_2_ × 2H_2_O
oxidizing agent	NaOH
capping agent	AA
particle stabilizer	C_6_H_7_NaO_7_
surfactant	PVP	PF127	SDBS	EG	PVA
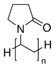	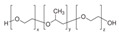	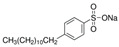	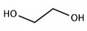	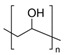

**Table 2 sensors-22-04783-t002:** The facet area and area ratio of copper oxide polyhedral materials. Calculation based on the SEM images.

Sample	Surface Composition	Facet Area (µm)^2^ on Single Particle				The Area Ratio of Facet on the Single Particle (%)				The Intensity Ratio of Cu_2_O (111) and Cu_2_O (200) * γ_(111)/(200)_
		(111)	(110)	(100)	(011)	(111)	(110)	(100)	(011)	
CC1	Cu_2_O/CuO	0.043	0.139	0.131	0.098	13.93	22.51	31.82	31.71	3.03
CC2			0.043	0.066	0.018		24.16	55.62	20.22	1.75
CC3		0.012	0.043	0.102	0.038	8.11	14.53	51.69	25.68	2.91
CC4		0.017	0.038	0.235	0.029	7.05	7.88	73.06	12.02	3.07
CC5		0.240				100				3.10

* The intensity ratios γ of copper oxide (200) and (111) planes may be evidence of the higher sensing/electrocatalytic activity due to the shifting toward lower potential values of initial glucose oxidation.

**Table 3 sensors-22-04783-t003:** Summary of the electrode electrochemical performance.

Sample	Electrochemical Active Surface Area				Slope	R^2^	Oxidation Potential [mV]	LOD	LOQ	Sensitivity [µAmM^−1^ mm^2^]	
	Slope	R^2^	EAS [mm^2^]							by: EAS	by: GCE (0.2826 mm^2^)
CC1	7.86 × 10^−5^	0.9994	10.89	AMP * CV *	1.45 5.60	0.9842 0.9910	660	0.30 0.16	0.90 0.48	0.13 0.51	5.13 19.82
CC2	6.01 × 10^−5^	0.9680	8.33	AMP CV	0.88 4.95	0.9286 0.9831	660	0.53 0.16	1.60 0.49	0.11 0.59	3.13 17.53
CC3	3.05 × 10^−5^	0.9823	4.22	AMP CV	45.00 5.47	0.9788 0.9988	660	0.20 0.07	0.60 0.20	10.66 1.30	159.24 19.36
CC4				AMP	13.51	0.9966		0.07	0.21	1.04	47.80
	9.39 × 10^−5^	0.9995	13.01	CV_an_ CV_cat_	9.66 8.55	0.9645 0.9906	650	0.11 0.11	0.72 0.35	0.77 0.66	34.19 30.26
CC5	-	-	-	AMP CV_an_ CV_cat_	12.57 14.65 16.40	0.9773 0.9854 0.9278	640	0.18 0.11 0.33	0.54 0.34 0.99	-	44.49 51.83 58.04

* CV—data obtained from cyclic voltammetry, index: an—anodic curve, cat—cathodic curve, AMP—data obtained from amperometric measurements.

## Data Availability

Not applicable.

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
