# Peer review of "Voltammetric Detection of Glucose—The Electrochemical Behavior of the Copper Oxide Materials with Well-Defined Facets"

_sensors, 2022, doi:10.3390/s22134783_

Round 1

Reviewer 1 Report

Since then, abundant literatures about copper oxide and cuprous oxide based non-enzymatic glucose sensors have been reported. The investigations have been focused on improving the analytical performances of the determination such as sensitivity, linear range, limit of detection and selectivity, while the facet-dependent Cu2O electrocatalysis has not been not fully investigated. In this context, the submitted work provides valuable information for the optimization of the copper oxide-based non enzymatic sensors for glucose determinations.

The work can be improved, taking into consideration the following suggestions and observations:

  1. Please, modify the title as follows: Voltammetric detection of glucose – the electrochemical behavior of the copper oxide materials with well-defined facets
  2. Please, explain the nature of the two reduction processes recorded at the voltammograms (p. 9, line 258).
  3. Please, provide the amperometric curves recorded in addition to Figs. 7 and 8.
  4. Please, provide data associated with the reproducibility of the determinations (as the electrode coverage in not homogeneous).

Author Response

The response to the Reviewer is included in the file below.

Reviewer 2 Report

The authors studied the electrochemical behavior of the copper oxide materials for the glucose detection. The characterization of the Cu2O was detailed, and the morphology of different materials was interesting. However, some issues, especially the performance of the glucose detection were misunderstanding. Therefore, I recommended this manuscript for publication after major revisions:

  1. How could the authors obtain the conclusions that “The chronoamperometric analyzes (AMP) exhibit a decrease in signal in contrast to the cyclic voltammetry data (CV), indicating that the number of active sites involved in glucose oxidation processes resulting from the structure of the material is insufficient.”
  2. The keyword “Nafion” in the manuscript was improper.
  3. What’s the role of the CC1-Au for the glucose detection?
  4. To give readers a comprehensive research background of glucose detection, some related references should be cited in the introduction, for example: Biosens. Bioelectron., 2018, 104, 152; Biosens. Bioelectron., 2020, 165, 112336.

Author Response

(The authors gave the same response as above.)

Reviewer 3 Report

The study sent by Anna Kusior is of interest to Sensors readers. The experimental part is consistent and the data are well explained in the Discussions chapter. The figures are useful and the references are well chosen.

However, the practical utility of the sensor has not been demonstrated considering that no selectivity studies have been performed and no actual tests have been tested.

The author is asked to explain why these studies are missing from the manuscript.

Moreover, there are countless experimental approaches to glucose detection in the literature. The authors should introduce a critical discussion section in which to compare the developed sensor with the literature.

Minor observation: the manuscript must be read carefully and revised because there are errors in English expression and typos that must be eliminated.

Author Response

(The authors gave the same response as above.)

Reviewer 4 Report

The work reported herein aims at exploring the use of copper oxide materials with different crystal structure in the enzyme less detection of glucose.

The work is well presented, and the synthesis and characterization of the copper oxide structure is adequate and well described.

On the other end when it comes to the electrochemical characterization and the use of such materials for the detection of glucose the work is showing significant weaknesses that need to be addressed.

1)      The authors need to improve the introduction when it comes to enzyme less detection of glucose and use existing literature also to understand their finding.

2)      Voltametric characterization of the different electrodes (including GC coated with Nafion) in electrolyte solution (KCl) is missing. This makes very hard, in the ferrocyanide experiment, to identify the signal from the copper oxide, the redox of ferrycianide at these materials, and those at the exposed GC.

3)      CV in ferrocyanide at GC coated with Nafion are also needed.

4)      At which potential the amperometric measurement were performed? Example of relevant amperometric measurements should be presented. Differences in current values between CV and amperometry are expected due to the nature of the two measurements (e.g. different influence from the charging current).

5)       Explanation on the amperometric response lines 328-338 is hard to understand and quite much speculative. This need to be improved and eventually supported by references.

6)      From the results presented herein it seems that better response are obtained when Cu(II) oxide is presented. Can the authors elaborate on this?

Author Response

(The authors gave the same response as above.)

Round 2

Reviewer 2 Report

My comments and suggestions have been taken into account. In my opinion, this manuscript can be now accepted for publication.

Author Response

Thank you for helping to improve this article.

Reviewer 3 Report

The quality of the manuscript was improved after the review and the author responded to the comments made by the reviewers.
In this form, the study may be recommended for publication in the Sensors journal

Author Response

(The authors gave the same response as above.)

Reviewer 4 Report

The reviewer acknowledge the effort of the authors in improving the introduction and the experimental part (new results).

Nevertheless the description and discussion of the electrochemical results still require to be improved. 

Author Response

Thank you for putting up your review and comments. According to the reviewer's comments, the corrections were made and can be found in the manuscript marked in blue.